# The Influence of Air Pollution on Happiness and Willingness to Pay for Clean Air in the Bohai Rim Area of China

**DOI:** 10.3390/ijerph19095534

**Published:** 2022-05-03

**Authors:** Qianqian Liu, Guanpeng Dong, Wenzhong Zhang, Jiaming Li

**Affiliations:** 1School of Geography, Nanjing Normal University, Nanjing 210023, China; 09430@njnu.edu.cn; 2Jiangsu Center for Collaborative Innovation in Geographical Information Resource Development and Application, Nanjing 210023, China; 3Key Research Institute of Yellow River Civilization and Sustainable Development, Henan University, Kaifeng 475005, China; 4Collaborative Innovation Center on Yellow River Civilization Jointly Built by Henan Province and Ministry of Education, Henan University, Kaifeng 475005, China; 5Institute of Geographic Sciences and Natural Resources Research, Chinese Academy of Sciences, Beijing 100101, China; zhangwz@igsnrr.ac.cn (W.Z.); lijm@igsnrr.ac.cn (J.L.)

**Keywords:** well-being, air pollution, multi-level modeling, willingness to pay, Bohai Rim area, China

## Abstract

Air pollution imposes detrimental impacts on residents’ health and the general quality of life. Quantifying the influential mechanism of air pollution on residents’ happiness and the economic value brought by environmental quality improvement could provide a scientific basis for the construction of livable cities. This study estimated urban residents’ willingness to pay for air pollution abatement by modeling the spatial relationship between air quality and self-rated happiness with a Bayesian multi-level ordinal categorical response model. Using large-scale geo-referenced survey data, collected in the Bohai Rim area of China (including 43 cities), we found that a standard deviation decrease in the number of polluted days over a year was associated with about a 15 percent increase in the odds of reporting a higher degree of happiness, after controlling for a wide range of individual- and city-scale covariate effects. On average, urban residents in the Bohai Rim region were willing to pay roughly 1.42 percent of their average monthly household income for mitigating marginal reductions in air pollution, although great spatial variability was also presented. Together, we hoped that these results could provide solid empirical evidence for China’s regional environmental policies aiming to promote individuals’ well-being.

## 1. Introduction

Benefiting from the reform and opening-up policy, China’s rapid industrialization and urbanization have brought inconceivable economic development and improvement to living standards. Unfortunately, many grave challenges, including insufficient public services, soaring housing prices, declines in food safety, and environmental degradation also become increasingly evident [1,2,3]. All these negative influences may restrict the promotion of residents’ happiness. Air pollution, especially ambient fine particulate matter (PM_2.5_), has attracted substantial attention as a primary pollutant that plays a pivotal role in the deterioration of the quality of urban living environments.

In confronting the task of tackling serious air pollution and improving residents’ self-rated happiness [4], it is vital to understand how air pollution affects self-rated happiness and to establish reliable estimates of residents’ willingness to pay for pollution reduction [5,6]. Improved environmental quality could facilitate better life satisfaction through various channels, either it being by enhancing productivity or by reducing the chances of physical illness [7,8]. A large and growing body of literature has documented the effects of socio-demographics, including income, health, unemployment, education, social communication, gender, marital status and religious attitudes, on self-rated happiness [5,9,10,11]. Residents also pay more and more attention to the physical environment in which they live and it is an increasingly important factor affecting happiness [12].

Studies closely related to the present research focused on the function of environmental quality on self-rated happiness and indicated that environmental quality, such as air pollution, noise and climate, exert significant influences on life satisfaction [13,14]. Increasingly, air pollution, especially ambient fine particulate matter, becomes a global concern and is an issue that has drawn substantial attention. For instance, Welsch and Ferreira explored the relationship between air pollution and self-rated happiness in 10 European countries and found that air pollution exerted a statistically significant impact on the difference in subjective well-being between countries at different times [14]. Besides the objectively measured air pollution, perceived air pollution (i.e., respondents’ ratings on pollution levels) was also negatively associated with life satisfaction [7]. The study undertaken by Ambrey et al. confirmed the negative relationship between life satisfaction and PM_10_ once the surrounding PM_10_ concentration exceeded the health standards in the study area [15]. Similar results were found by Orru et al. [12], who recorded that a 1 μg/m^3^ increase in the annual concentration of PM_10_ was associated with a decrease of 0.017 in life satisfaction on a ten-point scale in a large-scale European social survey.

Given China’s present levels of air pollution and the much-increased public awareness of the hazardous air pollution effects in recent years, a large number of studies have been carried out, aiming to establish the links between air quality and self-rated happiness in China. It is notable that SO_2_ [16], particulate matter [17], air pollution [18], PM_2.5_ and PM_10_ [19,20], and SO_2_ and NO_2_ [21] have been shown to exert significant negative influences on self-rated happiness [5,6]. Furthermore, such literature has enriched our knowledge of the relationship between air quality and self-rated happiness on a national scale, or an individual city scale, such as in Beijing or Shanghai. However, extant research, in general, has not yet discussed the effect of air pollution on happiness at a regional scale, leaving potential heterogeneity effects between cities under-explored. Moreover, they tended to focus on individual socio-demographic and economic characteristics of self-rated happiness, with relatively few studies having attached importance to the function of city-level factors.

One of the crucial concerns pertaining to the importance of air pollution on subjective well-being is how individuals evaluate these effects. Subjective life satisfaction data have been rigorously employed to elicit environmental amenity values, such as air quality. However, economic value cannot be directly observed through market transactions [7,22,23,24]. We refer to Ferreira and Moro and Welsch and Ferreira for a comprehensive discussion on the theoretical grounds of this approach [14,23]. Based on environmental evaluation methods, a growing body of empirical studies have been conducted to quantify the marginal willingness to pay (WTP) for environmental quality. A survey of studies most relevant to the present study was provided in Appendix A
Table A1. It is useful to note, however, that these studies yield quite different estimates on WTP for air quality. Such discrepancies could reflect true preference heterogeneities in WTP for clear air in different regions for reasons that are usually impossible or difficult for researchers to measure, or could be simply due to model misspecifications [14]. Either way, it may suggest that there is a need for more empirical pieces of evidence to better understand regional variability in WTP for air quality.

Drawing upon previous theories of happiness and empirical studies [20,23,25,26,27], we proposed an empirical framework to investigate the impact of air pollution on happiness, including both individual- and city-level factors, as displayed in Figure 1. The contribution of this paper is in two aspects. First, it seeks to integrate both individual attributes and city-level factors to access the linkages between air pollution and happiness at a regional scale. Second, this study adopts a multi-level ordinal category response model to simultaneously cope with the hierarchical structure of our survey data (respondents nesting into cities) and the ordinal self-rated happiness scores. By doing so, we proposed a multi-level model-based environmental evaluation method, producing not only a population estimate on WTP for air quality improvement for the study region but also its nuanced spatial variabilities across cities.

## 2. Methodology and Data

### 2.1. Statistical Model

Considering the two-level structure of our survey data where individuals nested into cities and the ordinal nature of the happiness measure, we adopted a multi-level ordinal response model in this study, where *y_ik,j_* denotes the self-rated happiness score of individual *i* living in city *k*. The cumulative probability of a happiness score falling in the *j*-th category or below *P*(*y_ik,j_* ≤ *j*) is linked to a series of individual- and city-level independent variables through a logit link function, expressed as [28]:(1)$$\log\frac{P(y_{ik,j}\leq j)}{1-P(y_{ik,j}\leq j)}=\alpha_{j}-X^{\prime }{}_{ik}\beta -Z^{\prime }{}_{ik}\gamma -u_{k}$$
where *α**_j_* (*j* = 1, 2, …, *J* − 1) denotes the threshold parameter related to the cumulative distribution of the *j*th response category. *X* is the individual-level predictor and *Z* is the city-level predictor. Vectors *β* and *γ* are the regression coefficients to be estimated. The vector *u* absorbs unobservable city-level influences on *P*(*y_ik,j_* ≤ *j*), which are assumed to follow a normal distribution *N*(0, *σ_u_*^2^). The importance of the unobservable influence can be quantified by utilizing the variance partitioning coefficient *σ_u_*^2^/(*σ_u_*^2^ + π^2^/3) [28]. With respect to the interpretation of the regression coefficients, they can be interpreted on the odds scale or the log odds scale. On the log odds scale, the regression coefficient of a predictor variable (e.g., income) represents the effect on the latent continuous happiness variable from this variable; which is similar to the interpretation of the coefficients under a standard multi-level linear regression model [29]. The latent happiness variable is mapped to the observed categorical happiness scores based on the set of cut points *α**_j_*.

The Bayesian inference approach, and the Markov chain Monte Carlo (MCMC) method, were used to estimate the multi-level ordinal response model, primarily because frequentist estimation methods such as the maximum likelihood approach have been found to be unstable for variance parameters [30]. Equation (1), together with the prior distributions for each unknown model parameter, completes our model specification. Following Browne and Gelman et al., a multivariate normal distribution with large hyperprior variance parameters was specified for the regression coefficient parameters with an inverse Gamma distribution for *σ_u_*^2^ [30,31]. Equation (1) was implemented using the MLwiN software package [32]. Statistical inferences on the model parameters were based on three MCMC chains, each of which consisted of 50,000 iterations with a burn-in period of 30,000 to ensure that the samples were able to converge. Every tenth sample of these iterations was retained to reduce autocorrelation [32]. A few other values were set for hyperprior parameters to assess the potential sensitivity of the estimation results, and only marginal differences in the estimation results were found.

In order to derive estimates on WTP for air quality improvement in line with the environmental evaluation literature [14], we need to build the link between reported discrete self-rated happiness scores with an unobserved continuous latent variable *happiness**. Following Agresti [29], it can be formulated as:$$y_{ijk}=1\;\;\mathrm{if}-\infty <happiness_{ik}^{*}\leq a_{1}$$
(2)$$y_{ijk}=j\;\;\mathrm{if}\;a_{j-1}<happiness_{ik}^{*}\leq a_{j};j=2,\ldots ,\;J-1$$
$$y_{ijk}=J\;\;\mathrm{if}\;a_{j}<happiness_{ik}^{*}\leq +\infty .$$

Based on the latent variable *happiness**, we can derive the formula to calculate the economic valuation of air quality (i.e., the marginal WTP for air quality improvement) as,
(3)$$WTP=\frac{\partial happiness^{*}}{\partial pollution}/\frac{\partial happiness^{*}}{\partial income}=\frac{\beta_{happiness}}{\gamma_{pollution}}\times income$$
where, $\beta_{happiness}$ is the regression coefficient of air pollution; $\gamma_{pollution}$ denotes the regression coefficient of income. The ratio of the two first-order partial derivatives yields the marginal rate of substitution between income and air pollution, measuring the amount of money placed on a marginal reduction in air pollution levels while holding the level of latent happiness constant [14].

### 2.2. Study Area, Data and Variables

The Bohai Rim area sits on the top list of China’s regions with high air pollution levels. For instance, in 2014, among 190 monitored cities, 10 cities with the highest air pollution were located in the Bohai Rim area. In addition, Xingtai city was the most severely polluted city with an annual average PM_2.5_ concentration being 131 μg/m^3^, whilst on about 30 occasions it exhibited a lasting pollution concentration of greater than 500 μg/m^3^ [33]. Our primary data was a relatively large-scale first-hand survey, collected from the Bohai Rim Urban Agglomeration (BRUA) from November 2014 to February 2015. The survey targeted urban residents who had lived in their current residences for at least six months, and excluded non-representative samples such as those who were temporary residents in the area on account of travel or business purposes. Details on the sampling strategy and the representativeness of samples were extensively discussed in Zhang et al. [34]. In total, 7500 questionnaires in 43 cities located within the BRUA were issued with about 6965 returned, of which 6552 questionnaires with completed information were used in this study. Regarding the sample size distribution, there were, on average, 152 respondents in each city with a range of between 82 and 208 respondents.

Happiness was the outcome variable in our study, measured by the survey question: ‘All things considered, how satisfied are you with your life as a whole?’. Responses were quantified on a 5-point Likert scale, ranging from 1 (very unhappy) to 5 (very happy). As shown in Table 1, more than two-thirds of the respondents were happy or very happy with their lives, whilst only about 4.3% of respondents rated their happiness level as low or very low. To reflect the spatial distribution of happiness, we mapped the percentages of respondents who reported that they were happy or very happy on the city scale (Figure 2). Spatial variations in happiness were clearly revealed: cities, including Dongying, Weifang, Liaoyang and Jinzhou were associated with the highest levels of happiness, followed by Zhangjiakou, Tieling, Fuxin, Rizhao, Linyi, Jining and Dezhou; residents living in Tianjin and Baoding appeared to have the lowest levels of happiness, on average.

The independent variables were categorized into two sets. On the city scale, the variable of primary interest was air pollution, measured by the number of polluted days when a city’s daily (24-h) averaged PM_2.5_ concentration exceeded 75 μg/m^3^, a pollution level that is detrimental to human health. What needs illustration is that the definition of polluted days is in accordance with China’s National Ambient Air Quality Standards. We acknowledge that the threshold of hazardous PM_2.5_ concentration level used in China was far higher than that used in Europe. PM_2.5_ concentration level data can be accessed at http://www.beijingcitylab.com/projects-1/13-pm2-5/, accessed on 1 December 2021. With this method, we calculated the number of polluted days for each city between April 2013 and April 2014. On average, about 42% of days were classified as polluted for all cities, although the variability (a standard deviation of 69 days) was also evident among cities in the region. Other city-scale independent variables included climatic factors (annual average wind speed, temperature and precipitation), economic development level (per capita GDP), and urban industrial structure measured by the shares of secondary industry in GDP. These variables were constructed by using data from 2013. As a robustness check, we also measured these independent variables by using their averages over the period from 2010 to 2014. Results from these averaged variables were similar to those reported below. On the individual scale, following previous literature [23], a range of socio-demographics, economic, health and housing characteristics variables were included in our happiness model. They included gender, age, education level, self-rated health conditions, marital status, housing tenure and type, and monthly income.

The initial income variable was coded as a categorical variable with seven bands. In order to produce an estimate on WTP, income was transferred to a continuous variable by utilizing the midpoint of each income band; and for the open-ended top category, by employing an extrapolation method [23]. Descriptions of the variables used in the analysis are provided in Table 1.

## 3. Empirical Findings

### 3.1. Influences of Air Pollution on Happiness

A series of multi-level ordinal response models were sequentially implemented. In Model 1, the income and air pollution variables were included; individual- and city-scale variables were added in Model 2; Model 3 further considered the potential impacts of the self-rated health status on happiness. Model estimation results were presented in Table 2. From Model 1, it could be seen that income was positively associated with the odds of reporting higher happiness levels, which was statistically significant at the 5% significance level. Air pollution, as anticipated, was negatively associated with happiness. To put such estimates into context, a standard deviation decrease in the number of polluted days over a year (i.e., 69 days) was associated with about a 15 percent increase in the odds of reporting a higher degree of happiness, everything else being equal. Such results, in general, echoed previous findings of studies investigating the environment-happiness relationship [6,7,20].

Adding individual socio-demographics and other city-level predictor variables into the happiness equation significantly improved the model fit, as indicated by a substantial decrease in DIC values from Model 1 to Model 2. It was, however, also noted that the coefficient of income almost halved from Model 1 to Model 2, after controlling for potential confounding factors. In contrast, the coefficient magnitude of air pollution doubled and remained statistically significantly negative. Such changes would have a large impact on the estimates of marginal WTP for air quality improvement, as is discussed below.

With respect to other city-level predictors, physical and climatic factors, including annual temperature, wind speed and precipitation, were not found to be statistically associated with individual happiness, after adjusting for potential compositional effects—the effects of individual-scale covariates [28]. Such results were, to some extent, contradictory to previous findings on the impacts of climatic variables on life satisfaction obtained based on European data [23]. This highlighted, in part, the consequences of methodology choices. For instance, physical environmental and climatic factors usually entered the life satisfaction or happiness equation as individual-level variables in previous studies [23], which led to the conflation of scales underlying the data and unreliable statistical inferences on regression coefficients. Moreover, there might also be an adaption effect—residents may have adapted to local physical and climatic factors, such as temperature and precipitation [2]. A rigorous test on such adaption effects would require a longitudinal design; this is beyond the scope of this study.

Local economic development (per capita GDP) was positively related to happiness, however, this relationship was not statistically significant, whilst local industrial structure was statistically significantly associated with happiness. China, as an industrializing economy, and the industrial structure, measured by shares of the second industry in GDP, was correlated with the local economic development of cities. The Pearson correlation coefficient between the two variables was about 0.5 in our data, which might explain the insignificance of local economic development on happiness. We also tried to test whether there was an interaction effect between individual income and local economic development; none was found.

With respect to individual socio-demographic factors, our estimation results were broadly in line with prior studies. Females, on average, tended to be happier than males. Married people reported higher levels of happiness compared to single or widowed persons ceteris paribus. Residents with higher levels of educational achievement (e.g., with a degree) tended to exhibit higher levels of happiness. In addition, homeowners, on average, were more likely to report higher happiness ratings, but housing type did not seem to affect happiness at a statistically significant level. This was indicated by the insignificant differences in the odds of reporting higher levels of happiness between people living in commodity housing and other types of housing.

In Model 3, we further included the self-rated health variables. This was intended to control for the effect of air pollution on health and obtain a more reliable estimate of the partial impact of air pollution on happiness. For instance, Ebenstein et al. have shown that air pollution has causal influences on life expectancy, based on a quasi-experimental design [35]. As shown in the model estimation results, self-rated health was statistically significant to happiness: better health was associated with higher levels of happiness. This finding corroborated those of previous studies. We did observe a slight decrease in the coefficient of air pollution after controlling for health effects. Nonetheless, the partial effect of air pollution on happiness was still statistically significant and negative. It is useful to note that the coefficient estimates for other covariates remained similar to those in Model 2.

### 3.2. Robustness Checks

We tested the robustness of the estimated relationship between air pollution and happiness; two alternative model specifications were analyzed. In the first model specification, we excluded the migrant samples from our data and implemented the same multi-level model as Model 2. The logic behind this was that migrants, identified by respondents with a local household registration (*hukou*), could be different from local urban residents in terms of evaluating environmental quality. The second robustness check had a methodological focus; the original five levels of happiness rating were collapsed into three categories by combining the responses of ‘very happy and ‘happy’ into one response and the responses of ‘very unhappy and ‘unhappy’ into another. The key reason for doing this was that the most extreme ratings of happiness (very unhappy and happy) only accounted for a small proportion of the responses. By making this change, the reliability of estimated multi-level ordered response statistical models could be increased in line with the views previously proffered by Agresti [29].

Robustness check results are presented in Table 3. In the first robustness check, air pollution and income were still statistically significantly associated with happiness at the 5% significance level. It seemed that local residents tended to be slightly more affected by air pollution, as indicated by an increase in the regression coefficient of air pollution. With respect to the second robustness check, the key results on the impacts of air pollution and income on happiness were similar to those discussed above. It is useful to note that the coefficient estimates on other predictor variables were very similar to those in Model 3, despite the trivial differences that were observed with regard to the magnitudes of the regression coefficients. Together, our robustness checks suggested that our key findings were relatively reliable.

### 3.3. Willingness to Pay by Bohai Rim Area Urban Residents

With our survey data, the average monthly income of urban families was calculated and presented in Figure 3. The average monthly income in the Bohai Rim region was 6704 yuan; the highest city-scale average monthly income was 10,810 yuan; the lowest city-scale average monthly income was 3314 yuan. There were 18 cities with a monthly average income (41.86% of the sample) higher than the average monthly income in the Bohai Rim region. These cities included Tianjin, Dalian, Zibo, Jinan, Qingdao, Dandong, and Zhangjiakou.

With model estimation results from Model 3, the marginal WTP for air quality improvement in the Bohai Rim region was roughly 95 RMB, equivalent to roughly 1.42 percent of the average monthly household income. This indicates that urban residents in the study area, on average, were willing to pay about 1.42 percent of their monthly income for a marginal decrease in air pollution—decreasing the number of polluted days by one. As a comparison, Dong et al. derived the marginal WTP for air quality improvement in the city of Beijing as roughly 30 percent of the average monthly income [36]. Their estimate was much higher than the one derived in this study, but still much lower than those reported in the European case studies [14]. However, great spatial variability in WTP for air quality improvement was also evident (Figure 3). For instance, residents in Tianjin city were willing to pay 225 RMB—2.27% of the average citywide monthly income—for a marginal reduction in air pollution. This presented the highest estimates on WTP for air quality improvement, followed by Dalian (202 RMB, 2.05%), Qingdao (198 RMB, 2.11%), Zibo (178 RMB, 1.88%), Laiwu (155 RMB, 1.86%), Yantai (155 RMB, 2.29%), and Jinan (151 RMB, 1.40%). Cities with low WTP for air quality improvement included Dongying, Chaoyang, Rizhao, Fushun and Jinzhou, as the average monthly income of these cities was relatively low.

Finally, for policy relevance, we further derived the monetary welfare effect of discrete changes, instead of a marginal change, in air pollution. For such a purpose, compensating surplus (CS) is a widely used measure in the environmental economics literature [14]. It was formulated as $\mathrm{CS}=Income[1-\exp\left(\frac{\beta_{happiness}}{\gamma_{pollution}}\times \Delta pollution\right)]$, in which Income was the mean income level and, Δpollution was the discrete amount of air pollution abatement. With model estimation results from Model 3, a reduction in the number of polluted days by a standard deviation (i.e., 69 days in the Bohai Rim region), the CS was calculated as about 4000 RMB (roughly 5.2 percent of the average household annual income). This could reflect a relatively strong will to reduce air pollution levels as people on average were willing to pay a fair amount of money for abating the number of days with heavy air pollution.

## 4. Conclusions and Implications

The last decade has seen a fast-growing interest in the environmental determinants of quality of life amongst academics and policymakers [12]. Using large-scale geo-coded survey data collected in the Bohai Rim region, this paper evaluated the association between air quality and happiness at a regional scale and investigated the willingness of residents to pay to alleviate air pollution as well as its spatial variability. A Bayesian multi-level ordinal response modeling framework was adopted to cope with the hierarchical structure of the survey data and the ordinal nature of the dependent variable. Some interesting conclusions were found. First, air pollution and income were statistically significantly related to self-rated happiness scores. A standard deviation decrease in the number of polluted days over a year was associated with about a 15 percent increase in the odds of reporting a higher degree of happiness, everything else being equal. Individual-level income was a stronger predictor of happiness, while the city-scale economic condition, measured by the per capita GDP, was not related to happiness. In addition, the marginal WTP for air quality improvement in the Bohai Rim region was roughly 1.42 percent of the average monthly household income, and residents were willing to pay roughly 5.2 percent of their average household annual income for reducing the number of polluted days by a standard deviation.

From this paper, we can obtain the following two enlightenments. First, the results indicated that residents in Tianjin, Jinan, Dalian, Qingdao and other cities with high-income levels have a relatively high willingness to pay, but residents in some cities with low-income levels also show a high willingness to pay, such as Yantai, Weihai, Tai’an, Anshan, Baoding and other cities with serious air pollution. It shows that residents’ willingness to pay is not only affected by household income levels but also affected by the air quality of the city residents live in. The results will help governments more accurately assess the investment income of environmental governance, and these quantitative results also provide a scientific basis for governments to formulate reasonable and targeted air pollution control policies. Second, with the increasing wealth of material life, Chinese residents begin to pay more attention to their own quality of life. Environment quality, as an important factor affecting residents’ happiness, plays an increasingly important role. At present, Chinese residents have a strong demand for a favorable environment, which means that the focus of urban construction should gradually shift from improving material living standards to improving the quality of life. The focus of urban governance should also shift from improving residents’ income to improving residents’ living environment quality and pursuing green growth and healthy growth.

Although the research in this paper supplements the relevant results of the relationship between environmental quality and residents’ happiness, limitations remain. First, due to the limitation of data acquisition, the year of data acquisition in this paper is 2014, and it is necessary to carry out updated data research in the future. In addition, only the PM_2.5_ pollution index was used to measure air pollution, and individual income was not measured using accurate actual income. Finally, due to the cross-sectional nature of our data, the associations between air pollution and self-rated happiness were not supposed to be interpreted as causal effects. With newly available survey data and quasi-experimental variations in air pollution dynamics generated by geographically varying environmental policies [35,37], the causal identification of how air pollution impacts individuals’ subjective well-being will be our future research priority.

## Figures and Tables

**Figure 1 ijerph-19-05534-f001:**
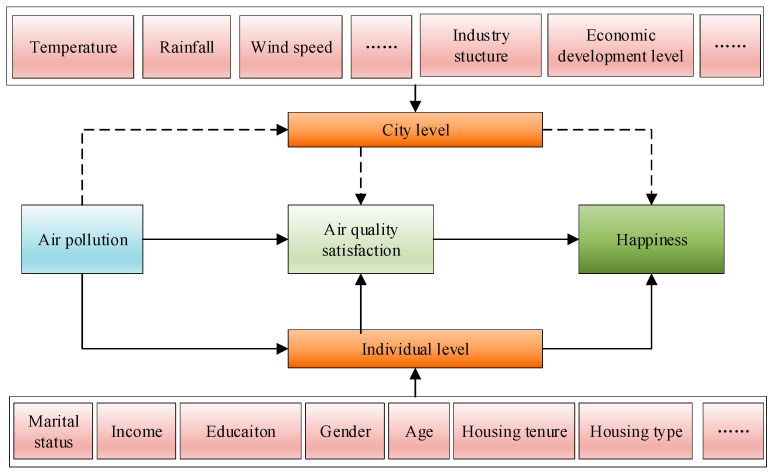
The conceptual framework of this study.

**Figure 2 ijerph-19-05534-f002:**
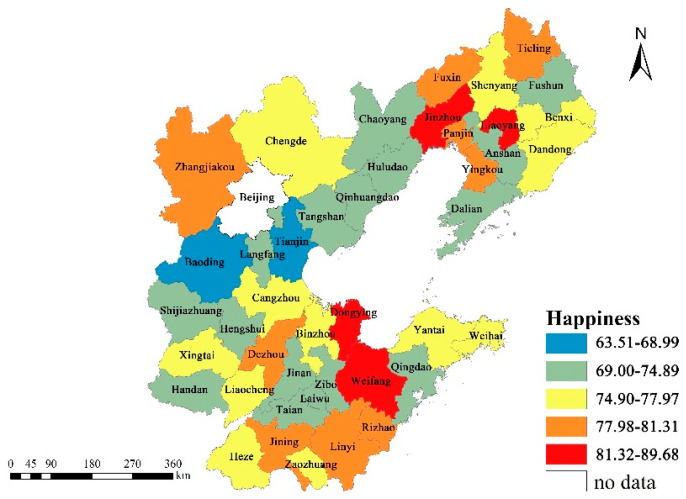
Spatial distribution of happiness in the Bohai Rim area.

**Figure 3 ijerph-19-05534-f003:**
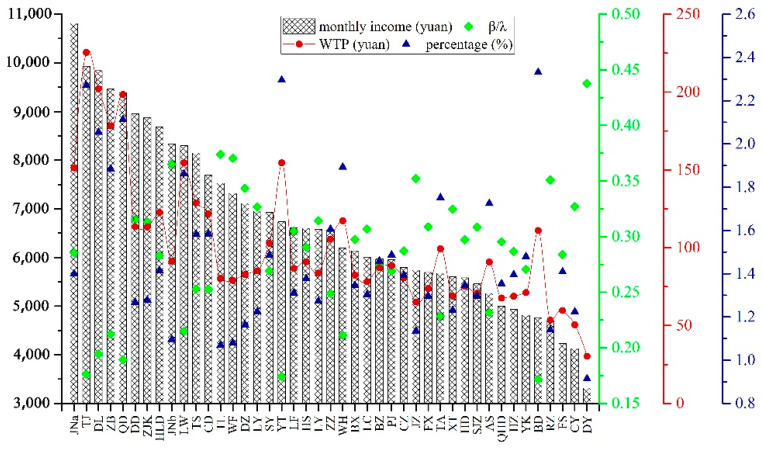
Willingness to pay and its proportion of monthly income from Bohai Rim Area urban residents. Those regions are coded by abbreviation as follows: JN^a^—Jinan, TJ—Tianjin, DL—Dalian, ZB—Zibo, QD—Qingdao, DD—Dandong, ZJK—Zhangjiakou, HLD—Huludao, JN^b^—Jining, LW—Laiwu, TS—Tangshan, CD—Chengde, TL—Tieling, WF—Weifang, DZ—Dezhou, LY—Linyi, SY—Shenyang, YT—Yantai, LF—Langfang, HS—Hengshui, LY—Liaoyang, ZZ—Zaozhuang, WH—Weihai, BX—Benxi, LC—Liaocheng, BZ—Binzhou, PJ—Panjin, CZ—Cangzhou, JZ—Jinzhou, FX—Fuxin, TA—Tai’an, XT—Xingtai, HD—Handan, SJZ—Shijiazhuang, AS—Anshan, QHD—Qinhuangdao, HZ—Heze, YK—Yingkou, BD—Baoding, RZ—Rizhao, FS—Fushun, CY—Chaoyang, and DY—Dongying.

**Table 1 ijerph-19-05534-t001:** Descriptive summary of variables in the analysis.

Variables	Description	Means (Standard Deviation)
Happiness	1 = very unhappy	1.11%
	2 = unhappy	3.24%
	3 = fair	26.76%
	4 = happy	51.18%
	5 = very happy	17.72%
Income	Monthly income (RMB)	6868 (5311)
Education	Junior college degree or above	62.04%
Housing type	Commodity housing	43.04%
Housing tenure	Owners	63.71%
Male	Female as base category	53.43%
Age	<20	3.68%
	20~30	39.74%
	30~40	29.53%
	40~50	25.21%
	>50	1.83%
Self-rated health	1 = very unsatisfied	0.49%
	2 = unsatisfied	4.17%
	3 = fair	27.47%
	4 = satisfied	48.43%
	5 = very satisfied	19.44%
Air pollution	Days with 24-h average The number of days with PM_2.5_ concentration exceeding 75 μg/m^3^	153 (69)
Wind speed	Average annual wind speed (m/s)	2.40 (0.27)
Temperature	Average annual temperature	12 (2.28)
Precipitation	Average annual precipitation (m^3^/h)	594 (115.13)
Per capita GDP	GPD/total population	60,217 (29,809)
Industry structure	Shares of secondary industry as % of GDP	46.69%

**Table 2 ijerph-19-05534-t002:** Model estimation results from multi-level ordinal response models.

Response	Model 1			Model 2			Model 3		
	Median	CI (2.5%)	CI (97.5%)	Median	CI (2.5%)	CI (97.5%)	Median	CI (2.5%)	CI (97.5%)
Income	0.504 ***	0.401	0.573	0.283 ***	0.198	0.357	0.284 ***	0.193	0.373
Air pollution	−0.002 **	−0.005	0.000	−0.004 **	−0.008	0.000	−0.003 ***	−0.006	−0.001
Commodity Housing				−0.06	−0.175	0.052	−0.067	−0.179	0.047
Owners				0.737	0.607	0.87	0.751 ***	0.623	0.881
Male				−0.159 ***	−0.256	−0.06	−0.179 ***	−0.277	−0.08
Age				−0.011	−0.082	0.057	0.001	−0.073	0.07
College degree				0.121 **	0.011	0.231	0.139 ***	0.031	0.247
Married				0.347 ***	0.208	0.487	0.310 ***	0.169	0.455
Wind speed				0.370 *	−0.216	1.108	0.564 ***	0.187	0.965
Temperature				0.057	−0.072	0.157	0.048 ***	−0.007	0.103
Industry structure				0.034 ***	0.005	0.079	0.024 ***	0.007	0.046
Per capita GDP				0.214	−0.51	1.000	−0.000 ***	0.000	0.000
Precipitation				−0.476	−2.141	0.905	0.000	−0.001	0.001
Self-rated health (reference: very bad)									
Bad							4.918 ***	5.76	4.102
Fair							3.122 ***	3.402	2.845
Good							2.243 ***	2.404	2.083
Very good							1.355 ***	1.496	1.214
City-level variance	0.003	0.002	0.006	0.004	0.003	0.008	0.113	0.056	0.21
DIC	14,558	-	-	13,354	-	-	13,280	-	-

Note: *, ** and *** represents a statistical significance level of 0.1, 0.05 and 0.01, respectively.

**Table 3 ijerph-19-05534-t003:** Robust check results with different model specifications.

Response	Robust I			Robust II		
	Median	CI (2.5%)	CI (97.5%)	Median	CI (2.5%)	CI (97.5%)
Income	0.285 ***	0.146	0.39	0.352 ***	0.284	0.456
Air pollution	−0.005 ***	−0.007	−0.001	−0.004 ***	−0.011	−0.001
Commodity housing	−0.044	−0.165	0.079	−0.031	−0.161	0.111
Owner	0.714 ***	0.572	0.86	0.790 ***	0.645	0.936
Male	−0.137 ***	−0.246	−0.03	−0.221 ***	−0.329	−0.11
Age	−0.052 *	−0.131	0.026	−0.034	−0.115	0.048
College degree	0.157 ***	0.038	0.28	0.251 ***	0.126	0.379
Married	0.353 ***	0.19	0.519	0.400 ***	0.247	0.554
Wind speed	0.378	−0.408	1.101	0.525 ***	0.24	0.974
Temperature	0.062 *	−0.03	0.149	0.005	−0.085	0.182
Industry structure	0.026 **	−0.002	0.059	0.050 ***	0.028	0.073
Per capita GDP	−0.000 **	0.000	0.000	−0.000 *	0.000	0.000
Percipitation	0.000	−0.001	0.002	0.000	−0.001	0.001
Self-rated health (reference: very bad)						
Bad	5.409 ***	−6.394	−4.453	3.773 ***	−4.534	−3.009
Fair	3.253 ***	−3.563	−2.943	2.77 ***	−3.066	−2.464
Good	2.331 ***	−2.509	−2.15	1.815 ***	−2.012	−1.628
Very good	1.435 ***	−1.592	−1.277	0.87 ***	−1.057	−0.684
City level variance	0.148	−0.275	−0.078	0.004	−0.007	−0.002
DIC	11,012	-		8474	-	

Note: *, **, and *** represent statistical significance levels of 0.1, 0.05, and 0.01, respectively.

## Data Availability

The data presented in this study are available on reasonable request from the corresponding author.

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
