# Peer review of "The Influence of Air Pollution on Happiness and Willingness to Pay for Clean Air in the Bohai Rim Area of China"

_ijerph, 2022, doi:10.3390/ijerph19095534_

Round 1

Reviewer 1 Report

Dear authors,

It is a great pleasure to peer review an interesting and well-written manuscript. Through the review process I’ve identified the following strengths and weakness.

The manuscript’s strengths:

  • The subject of the manuscript is interesting for both scientific community and governments;
  • The Introduction section provides a detailed presentation of the subject and what aimed the authors to conduct the research;
  • The methodology used is well and adequately described– an important aspect for the research community (other researches could very easy follow and adopt the methodology steps);
  • All figures properly show data and are easy to interpret and understand; authors interpreted appropriately and consistently the data throughout the manuscript;
  • The results are clearly presented and interpreted appropriately;
  • The conclusion drawn are supported by results;
  • The Reference section is consistent and include current cited references.

Regarding the weaknesses of the study, I don't have much to say. In the References chapter I identified that at no [35] is missing page numbers.

Reviewer 2 Report

This study analyzes the influence of air quality on the intention to pay. It is necessary to consider the following points.

First, since this paper based on individual-level survey data, it is necessary to examine the sampling method and representativeness issues. In 43 cities, 152 respondents in each city with a range of between 82 and 208. Differences in the number of samples between cities may affect the statistical results, and if representativeness of the sample is not secured, there will be a sampling bias problem.

Second, it is necessary to review the research literature on the relationships between air pollution and wtp, which is the focus of this study. There are various theoretical variables that can affect WTP. Authors should provide the theoretical backgrounds for whether the variables set in this study set these influencing factors appropriately is an important variable.

Third, is there any way to update the data? Since the data is from 2014, it is necessary to review the availability of the latest data. In addition, it is necessary to consider the problem of time lag between variables.

Reviewer 3 Report

I would like to thank the authors for this research that proposed an empirical framework to investigate the impact of air pollution on happiness, on both individual- and city-level. The new proposal tried to integrate both individual attributes and city-level factors to access the linkages between air pollution and happiness at a regional Chinese scale.

Also, the research adopts a multi-level ordinal category response model to simultaneously cope with the hierarchical structure of the survey data (respondents nesting into cities) and the ordinal self-rated happiness scores.

The proposed multi-level model-based environmental evaluation methods, produced a population estimates on WTP for air quality improvement for the study region. It also nuanced spatial variabilities across cities.

The research subject is timely, innovative, and highly interesting. It also fits the aim and scope of the journal.

The research is well designed and follows a sound scientific research method.  Results and recommendations are clear and could have an impact among the community of researchers.

However, some modifications are needed in order to improve the quality of the paper.

The title needs improvements. Estimating residents’ willing to pay for air pollution reduction is only one component out of the three components of the research. A more general and insightful title is needed.

The abstract needs improvements. What is the meaning of this sentences? “ Quantifying the economic value brought by relevant environmental policies that aims to promote 13 residents’ wellbeing”. It is disconnected from the previous one. You also need to mention which city the paper talks about.

You need to adjust key words. You can add these key words: well-being; air pollution; china

A number of recent references are suggested in order to improve the first part of the introduction.

Alsalama, T., Koç, M., & Isaifan, R. J. (2021). Mitigation of urban air pollution with green vegetation for sustainable cities: a review. International Journal of Global Warming, 25(3-4), 498-515.

Liu, Y., Zhu, K., Li, R. L., Song, Y., & Zhang, Z. J. (2021). Air Pollution Impairs Subjective Happiness by Damaging Their Health. International Journal of Environmental Research and Public Health, 18(19), 10319.

Liu, Y., Li, R. L., Song, Y., & Zhang, Z. J. (2019). The role of environmental tax in alleviating the impact of environmental pollution on residents’ happiness in China. International Journal of Environmental Research and Public Health, 16(22), 4574.

Kooli, C., & Muftah, H. A. (2020). Impact of the legal context on protecting and guaranteeing women’s rights at work in the MENA region. Journal of International Women's Studies, 21(6), 98-121.

Kooli, C. (2021). COVID-19: Public health issues and ethical dilemmas. Ethics, Medicine and Public Health, 17, 100635.

Lines 356-35: You said: “this paper, to the best of our knowledge, for the first time evaluated the association between air quality and happiness at a regional scale, and investigated the willingness of residents to pay to al- 358 leviate air pollution as well as its spatial variability.”

I do not agree with this statement. I invite you check these two research papers related to china. I also, highly encourage you to include these two papers in your discussion. You need to compare between results.

Liu, Y., Zhu, K., Li, R. L., Song, Y., & Zhang, Z. J. (2021). Air Pollution Impairs Subjective Happiness by Damaging Their Health. International Journal of Environmental Research and Public Health, 18(19), 10319.

Liu, Y., Li, R. L., Song, Y., & Zhang, Z. J. (2019). The role of environmental tax in alleviating the impact of environmental pollution on residents’ happiness in China. International Journal of Environmental Research and Public Health, 16(22), 4574.

Other minor comments are directly attached to the manuscript.

Round 2

Reviewer 2 Report

I did not find any reasons change the first decision

Reviewer 3 Report

I would like to thank the authors for this research that proposed an empirical framework to investigate the impact of air pollution on happiness, on both individual- and city-level. The new proposal tried to integrate both individual attributes and city-level factors to access the linkages between air pollution and happiness at a regional Chinese scale.

Also, the research adopts a multi-level ordinal category response model to simultaneously cope with the hierarchical structure of the survey data (respondents nesting into cities) and the ordinal self-rated happiness scores.

The proposed multi-level model-based environmental evaluation methods, produced a population estimates on WTP for air quality improvement for the study region. It also nuanced spatial variabilities across cities.

The research subject is timely, innovative, and highly interesting. It also fits the aim and scope of the journal.

The research is well designed and follows a sound scientific research method.  Results and recommendations are clear and could have an impact among the community of researchers.

Authors made the necessary suggested modifications. The manuscript reached an optimal level.